# Characteristics of new HIV diagnoses over 1995–2019: A clinic-based study in Montréal, Canada

Katia Giguère[1]*, Maliheh Vaziri[2], Clément Olivier[2], Louise Charest[2], Jason Szabo[2], Réjean Thomas[2], Mathieu Maheu-Giroux[3]

1 Centre de Recherche du CHUM, Université de Montréal, Montréal, QC, Canada, 2 Clinique médicale l'Actuel, Montréal, QC, Canada, 3 Department of Epidemiology and Biostatistics, School of Population and Global Health, McGill University, Montréal, QC, Canada

* katia.giguere.1@gmail.com

## Abstract

### Background

Characterization of populations at risk of acquiring HIV is required to inform the public health response to HIV. To identify potential changing needs in HIV prevention and care cascade, we aim to describe how the demographic profiles and exposure categories of newly diagnosed HIV positive individuals attending a large sexual health clinic in Montréal (Canada) evolved since the beginning of the antiretroviral therapy era in the mid-1990s.

### Methods

Using diagnosis data from participants of the *Clinique médicale l'Actuel* cohort of HIV-positive patients, we examined the distribution of exposure categories (sexual orientation, sexual behaviours, injection drug use, being born in an HIV-endemic country) by gender and year of diagnosis. Time trends in mean age and in the proportion of patients with late (CD4 <350 cells/μL) or advanced stage (CD4 <200 cells/μL) of HIV infection at diagnosis were assessed through meta-regressions.

### Results

A total of 2,612 patients diagnosed with HIV between January 1st, 1995 and December 31st, 2019 were included. Overall, mean age was 35 years (standard deviation: 10 years) and remained stable over time. The proportion of patients with advanced stage of HIV infection decreased from 16% in 1995 to 4% in 2019. Although men who have sex with men (MSM) consistently accounted for the highest proportion of new diagnoses (77%, 2,022/2,612 overall), their proportion decreased since 2013. There was also a concomitant decrease in the proportion of people who inject drugs, with none of the newly diagnosed participants reporting injection drug use since 2017, and an important increase in the proportion of patients born in an HIV-endemic country (24%, 7/29 in 2019), especially among women. Compared to patients from non-endemic countries, those from HIV-endemic countries were characterized by higher proportions of heterosexuals (88% vs 17%) and of women (52% vs 7%), and were twice likely to get diagnosed at an advanced stage of HIV infection (32% vs 15%).

**Data Availability Statement:** Data cannot be shared publicly because they contain potentially identifying or sensitive patient information. The data underlying the results presented in the study

are available from Clinique médicale l'Actuel at: info@lactuel.ca (https://cliniquelactuel.com/).

**Funding:** This work was supported by a postdoctoral award through Fonds de recherche du Québec - Santé (FRQS: http://www.frqs.gouv.qc.ca/en/) to KG, and through a Canada Research Chair in Population Health Modeling and grants from the Canadian Foundation for AIDS Research (CANFAR: https://canfar.com/) and the Canadian Institutes of Health Research (CIHR: https://cihr-irsc.gc.ca/e/193.html) to MMG. The funders had no role in study design, data collection and analysis, decision to publish, or preparation of the manuscript.

**Competing interests:** I have read the journal's policy and the authors of this manuscript have the following competing interests: KG reports postdoctoral awardsfrom the Fonds de recherche du Québec - Santé, the Canadian Institutes of Health Research (176645), and the CIHR Canadian HIV Trials Network during the conduct of the study, and personal fees from UNAIDS, outside the submitted work; MMG's research program is supported by a Canada Research Chair (Tier 2) in Population Health Modeling, and grants from the Canadian Foundation for AIDS Research and the Canadian Institutes of Health Research, and funding from UNAIDS and WHO, outside the submitted work. This does not alter our adherence to PLOS ONE policies on sharing data and material.

## Conclusions

In absolute numbers, MSM continue to account for the largest exposure category. However, patients from HIV-endemic countries, who tend to be diagnosed at later stages of HIV infection, constitute an increasing proportion of newly diagnosed individuals. These persons could face distinct barriers to rapid diagnosis. Tailoring HIV testing strategies and other prevention interventions to the specific unmet prevention needs of these individuals is warranted.

## Introduction

Forty years after the first reported case of acquired immunodeficiency syndrome (AIDS), human immunodeficiency virus (HIV) remains a global public health issue [1]. In 2019, a total of 2,122 new HIV diagnoses were reported in Canada, of which 646 (30%) were located in the province of Québec [2, 3]. In Québec, the burden is concentrated in Montréal with 9.1 new HIV diagnoses per 100,000 person-year in 2019 [3].

HIV diagnosis is the necessary first step in the HIV care cascade. Early diagnosis and treatment drastically reduce morbidities and improve survival among people living with HIV (PLHIV) [4]. Viral load suppression among treated PLHIV also prevents onwards transmission at population level [5]. Achievement of early diagnosis requires effective HIV testing services. Program data on the number of new HIV diagnoses also form the bedrock of quality HIV surveillance programs [6].

In Québec, HIV prevention is integrated into a comprehensive strategy for preventing sexually transmitted and blood-borne infections (STBBI) [7]. Over 1992–1997, the number of anonymous HIV testing centers was also gradually increased to cover all regions of the province in order to improve access to HIV testing and public health awareness [8, 9]. Most of HIV testing services, including HIV testing, pre- and post-test counselling, and linkage to appropriate HIV prevention, treatment and care services, were provided in free local community service centers and clinics that were run and maintained by the provincial government [9]. In 2004, the Québec *Ministry of Health and Social Services* called for more targeted screening approaches to increase HIV testing among the most vulnerable groups. These groups include: gay, bisexual, and other men who have sex with men (MSM), people who inject drugs (PWID), individuals born in an HIV-endemic country (e.g. Haïti), incarcerated individuals, youth in difficulty, female sex workers, and indigenous individuals [10].

In 2013, the World Health Organization recommended earlier initiation of antiretroviral therapy at a CD4 count ≤500 cells/μL for all adults and children above 5 years [11]. The same year, the *Quebec Ministry of Health* published its first interim guidance on pre-exposure prophylaxis (PrEP), stating that tenofovir/emtricitabine use could be beneficial to prevent HIV infection among seronegative MSM and serodiscordant couples [12], and in Montréal, the *Clinique médicale l'Actuel* (hereafter, *l'Actuel*), a large health center dedicated to the screening and treatment of STBBI since 1987, opened the first PrEP clinic in Canada, fostering access to and PrEP uptake, especially among MSM [13–15].

In 2017, Montréal became the first Canadian *Fast-Track City* by signing the *Paris Declaration*, on fast-track to end AIDS by 2030 [16]. This ambitious commitment implies that the HIV care cascade be strengthened to ensure that 90% of PLHIV be diagnosed, 90% of diagnosed PLHIV be on treatment, and 90% of treated PLHIV be virally suppressed [16]. With

different vulnerable populations facing diverse HIV prevention challenges, it is imperative to properly characterize the most affected populations in order to inform the public health response to HIV. Using 25 years of clinical data from *l'Actuel*, we aim to describe how the demographic profiles and exposure categories of newly diagnosed evolved over time since the beginning of the antiretroviral therapy (ART) era in mid-1990s. As the city contemplates how best to achieve HIV elimination, understanding shifts in the epidemiology of new diagnoses is warranted and will help identify unmet HIV prevention needs.

## Methods

### Setting

In 1999, *l'Actuel* established an open cohort that prospectively enrolled newly diagnosed HIV-positive individuals and retrospectively enrolled HIV-positive patients who had been diagnosed before 1999 (either at the clinic or outside the clinic) to study their demographic, behavioural, and clinical characteristics. In-person recruitment of participants is ongoing since 1999. All newly diagnosed HIV-positive adults (≥18 years) at *l'Actuel* are invited by a healthcare provider, in a second visit, to participate to the cohort. HIV-positive adults seeking care for the first time at *l'Actuel* since having been previously diagnosed somewhere else are also invited to participate. At the cohort's inception, HIV-positive patients who were already followed at the clinic were invited to participate to the cohort at their first visit following its establishment. All participants provided a free and informed written consent before recruitment.

### Participants

All HIV-positive patients who provided consent to participate to the cohort and had a known HIV-diagnosis date were eligible for inclusion in this study.

### Variables and measurements

Self-reported date of birth, gender (men, women, transgender), ethno-cultural background, sexual orientation (heterosexual, homosexual, bisexual), and HIV exposure categories were assessed through face-to-face interview by a healthcare provider at enrolment. Exposure categories are not mutually exclusive and include sexual orientation (among men), sexual behaviours (condomless sex, having a partner at risk of HIV, having an HIV-positive partner, having multiple sexual partners, and having ever engaged in sex work), an history of injection drug use, being born in an HIV-endemic country (including Haïti, and sub-Saharan African and Caribbean countries), mother-to-child transmission, and accidental exposure (i.e., injury with a contaminated object).

For patients diagnosed at *l'Actuel*, the HIV-diagnosis date corresponds to the compiled date of the first HIV-positive test to the patient's medical record at the clinic. For patients who were diagnosed before their first visit to *l'Actuel*, the date of the first HIV-positive test (diagnosis date) was self-reported. These patients were confirmed positive for HIV at the clinic before being enrolled in the cohort.

Age at diagnosis was calculated by subtracting the date of birth from the date of HIV diagnosis. CD4 count at HIV diagnosis was not available for patients diagnosed outside of *l'Actuel* clinic and, because this laboratory test was not widely used before the 1990s, for some patients diagnosed at the clinic. For diagnoses without a CD4 count measurement, we used the first CD4 count performed at the clinic–if measured within one year of diagnosis–to limit the number of missing observations.

## Statistical analyses

Authors had full access to the de-identified and anonymized database population. Related variables were cross-validated in order to identify potential data misclassification and, if warranted, corrected by verifying the patients' medical records.

Because we included both patients diagnosed at *l'Actuel* and patients initiating care at the clinic but diagnosed elsewhere, we expected that the last few years of data would be affected by right truncation, especially in 2020 due to the COVID-19 global pandemic. In addition, because patients diagnosed before the beginning of ART era, in mid-1990's, could have passed away before recruitment to the cohort in 1999, left truncation could be an issue. To avoid these, analyses were restricted to patients diagnosed between 1995 and 2019.

We used descriptive analyses to characterize HIV-positive patients at diagnosis, overall and stratifying by gender (men, women) and origin (being born in an HIV-endemic country or not). We examined gender- and age-specific temporal trends along with the distribution of exposure categories (sexual orientation, sexual behaviours, history of injection drug use, origin). Time trends in the mean age and in the proportion of patients with late (CD4 cells count $<350$ cells/μL) or advanced stage of HIV infection (CD4 cells count $<200$ cells/μL) at diagnosis were assessed through meta-regressions, using calendar year as a continuous independent variable (linear). We used t-tests to evaluate the statistical significance, and the coefficient of determination ($R^2$) to quantify the proportion of the variance explained by the independent variable. To assess the potential selection bias due to missing data for the CD4 count at diagnosis, we compared the characteristics at diagnosis between participants with and without CD4 count at diagnosis in a post-hoc sensitivity analysis. We used R 4.0.2 for all analyses and the R *meta* package for meta-regressions. This study was reported according to the *REporting of studies Conducted user Observational Routinely-collected health Data* (RECORD) statement guidelines (S1 Table).

## Ethics approval

All analyses were performed on de-identified and anonymized data from participants who consented participation in the study. This study was approved by the Veritas Institutional and McGill University's Faculty of Medicine Institutional Review Boards (A12-E84-18A).

## Results

A total of 5,554 HIV-positive patients were invited to participate to the *l'Actuel*'s HIV-positive cohort between January 1999 and March 2020 (Fig 1). Of these patients, 3,502 (63%) consented to participate and were recruited to the cohort. Forty two patients with an unknown date of diagnosis, 845 patients diagnosed before 1995, and 3 patients diagnosed in 2020 were excluded from analyses. A total of 2,612 participants diagnosed between January 1st, 1995 and December 31, 2019 were analysed in this study. The distribution of new diagnoses is presented in Fig 2.

Characteristics of participants are presented by gender in Table 1. One participant who reported being transwoman, homosexual, and to have sex with men was included among women in all tables. Newly diagnosed participants were mostly MSM (homosexual or bisexual men, 77%, 2,022/2,612). Overall, exposure routes through sexual behaviours were highly prevalent (88%, 2,293/2,612). Among men and women, the most common exposures were condomless sex (59%, 1,384/2,342) and having a partner at risk of HIV (40%, 107/270), respectively. The proportions of women reporting an history of injection drug use (19%, 52/270) and being born in an HIV-endemic country (42%, 113/270) were, respectively, over two and eight times that observed among men. We observed no differences between the median

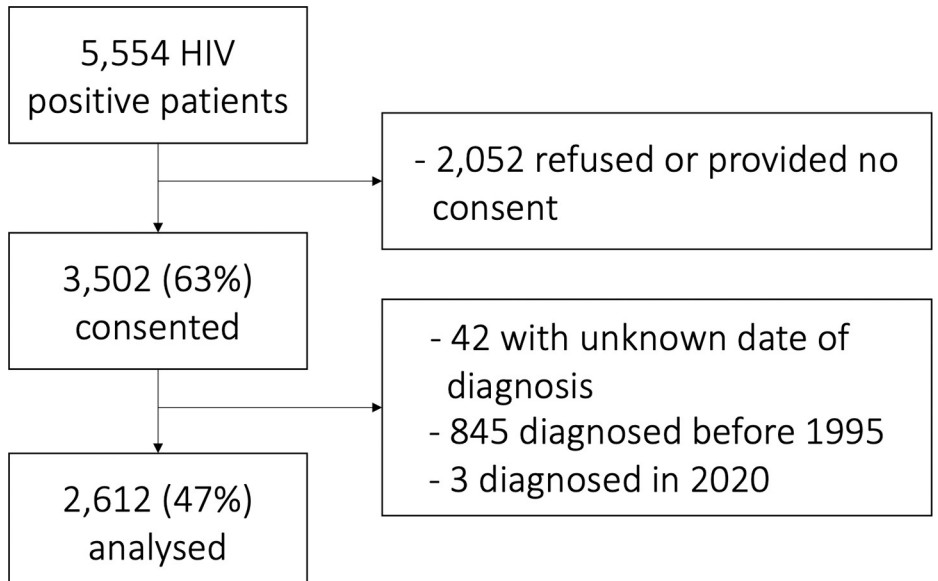

**Fig 1. Flow diagram of participants to the HIV-positive cohort at the *Clinique médicale l'Actuel* in Montréal, Canada.**

CD4 count at diagnosis among men (440 cells/µL, interquartile range [IQR] = 280–620 cells/µL, n = 1,737) and women (420 cells/µL, IQR = 225–580 cells/µL, n = 153). However, higher proportions of women than men were diagnosed at a CD4 count <350 cells/µL (39% vs 35%) or at a CD4 count <200 cells/µL (20% vs 15%).

From 1995 to 2019, the overall mean age at diagnosis was 35 years (standard deviation [SD]: 10 years) and remained stable over time (p-trend = 0.44, $R^2$ = 5%). Statistically significant positive trends in mean age at diagnosis were observed among women (Fig 3A, p-trend = 0.04, $R^2$ = 25%), heterosexual men (p-trend<0.001, $R^2$ = 100%), and PWID (p-trend = <0.001,

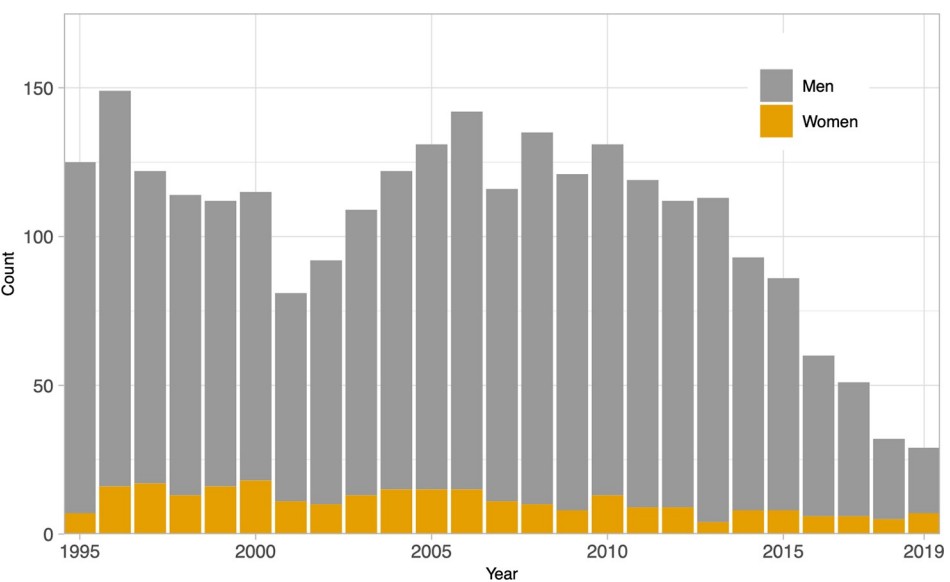

**Fig 2. Distribution of new HIV diagnoses by gender and year, 1995–2019 (n = 2,612).**

**Table 1. Characteristics of patients at HIV diagnosis among participants to the HIV-positive cohort at the *Clinique médicale l'Actuel* in Montréal (Canada), stratified by gender (n = 2,612), 1995–2019.**

| Characteristics | Overall | | Men | | Women [a] | |
| --- | --- | --- | --- | --- | --- | --- |
| | n = 2,612 | | n = 2,342 | | n = 270 | |
| | n | (%) | n | (%) | n | (%) |
| Mean age, years (SD) | 35 | (10) | 36 | (10) | 33 | (10) |
| Ethno-cultural background | | | | | | |
| Caucasian | 960 | (37) | 909 | (39) | 51 | (19) |
| Black | 244 | (9) | 129 | (6) | 115 | (43) |
| Hispanic | 183 | (7) | 175 | (8) | 8 | (3) |
| Asian | 66 | (3) | 65 | (3) | 1 | (0.4) |
| Native | 9 | (0.3) | 8 | (0.3) | 1 | (0.4) |
| Other | 51 | (2) | 48 | (2) | 3 | (1) |
| Unknown | 1,099 | (42) | 1,008 | (43) | 91 | (34) |
| Sexual orientation | | | | | | |
| Homosexual | 1,876 | (72) | 1,874 | (80) | 2 | (1) |
| Heterosexual | 590 | (23) | 328 | (14) | 262 | (97) |
| Bisexual | 146 | (6) | 140 | (6) | 6 | (2) |
| Exposure category [b] | | | | | | |
| Sexual behaviours | 2,293 | (88) | 2,129 | (91) | 164 | (61) |
| Condomless sex | 1,449 | (56) | 1,384 | (59) | 65 | (24) |
| Having a partner at risk of HIV [c] | 784 | (30) | 677 | (29) | 107 | (40) |
| Having an HIV-positive partner | 506 | (19) | 438 | (19) | 68 | (25) |
| Having multiple sexual partners (>1) | 680 | (26) | 663 | (28) | 17 | (6) |
| Having ever engaged in sex work | 73 | (3) | 50 | (2) | 23 | (9) |
| Injection drug use | 271 | (10) | 219 | (9) | 52 | (19) |
| Born in an HIV-endemic country | 219 | (8) | 106 | (5) | 113 | (42) |
| Contaminated blood transfusion | 6 | (0.2) | 4 | (0.2) | 2 | (1) |
| Other [d] | 10 | (0.4) | 8 | (0.3) | 2 | (1) |
| CD4 count, cells/μL (n = 1,890) | | | | | | |
| Median (IQR) | 440 | (280–620) | 440 | (280–620) | 420 | (225–580) |
| ≥500 | 779 | (41) | 722 | (42) | 57 | (37) |
| 350–499 | 447 | (24) | 411 | (24) | 36 | (24) |
| 200–349 | 369 | (20) | 340 | (20) | 29 | (19) |
| 50–199 | 223 | (12) | 198 | (11) | 25 | (16) |
| <50 | 72 | (4) | 66 | (4) | 6 | (4) |

SD: Standard deviation, IQR: interquartile range.

[a] Women include one transwoman who reported being homosexual and to have sex with men.

[b] Categories are not mutually exclusive.

[c] Partners at risk of HIV include people who inject drugs, HIV-positive individuals, individuals born in an HIV-endemic country, bisexuals, sex workers, and hemophiliacs.

[d] Other includes accidental exposure and mother-to-child transmission.

$R^2$ = 100%) (S1 Fig). The proportion of newly diagnosed individuals aged ≥45 years increased over time, whereas this age group accounted for 15% of new diagnoses before 2009, but for more than 20% of yearly new diagnoses from 2009 onwards (Fig 3B).

The relative distribution of exposure categories shifted since 1995 (Fig 4). Among men, although MSM always accounted for the highest proportion of new diagnoses, the proportion of heterosexual men increased since 2013, reaching 18% (4/22) in 2019 (Fig 4A). While

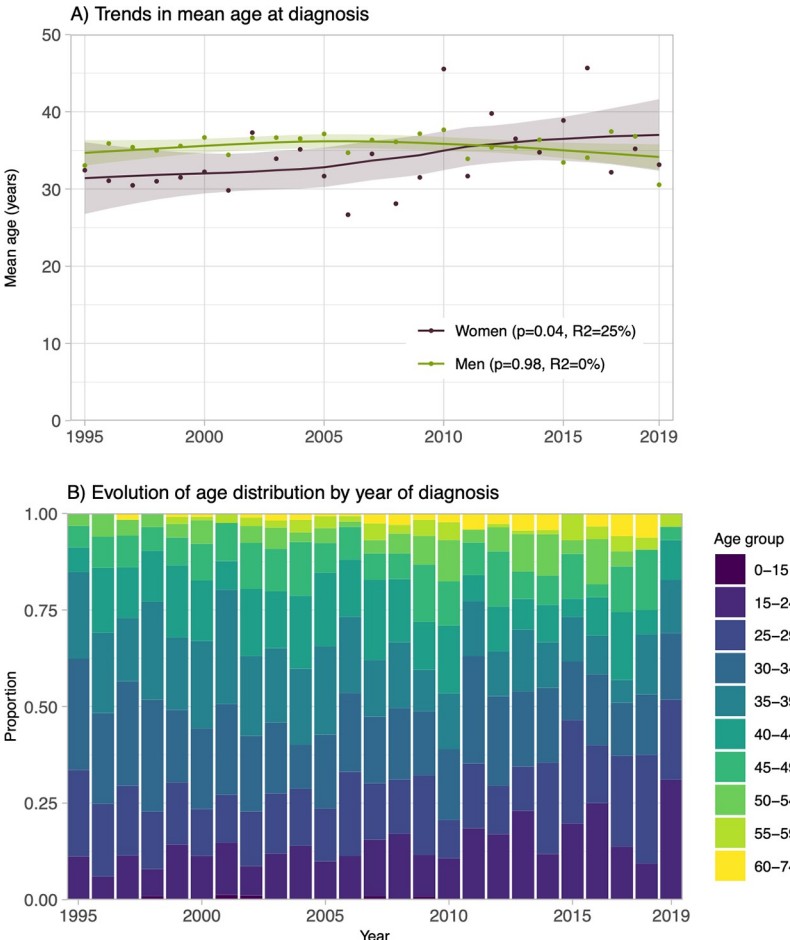

**Fig 3. Evolution of age at HIV diagnosis among participants to the HIV-positive cohort at the *Clinique médicale l'Actuel* in Montréal (Canada) over 1995–2019 (n = 2,612).** Panel A displays the mean age at diagnosis stratified by gender for each calendar year. Trends are displayed using local linear fitting and a degree of smoothing of 0.75. Linear time trends (p-values) were assessed by the mean of t-tests within univariate meta-regressions where the year of diagnosis was input as a continuous predictor variable. Panel B shows the underlying age distributions, categorized in ten groups, for both genders combined.

exposure through sexual behaviours remained highly prevalent (>75%) over time among men (Fig 4B left), their prevalence tended to vary among women, but have remained under 50% since 2016 (Fig 4B right). Among both men and women, there was a decrease in the proportion of PWID and, since 2017, none of the newly diagnosed participants reported injection drug use (Fig 4C). Although the yearly absolute number of newly diagnosed participants born in an HIV-endemic country did not seem to increase since late 1990's and remained under 15, there was an important increase in their proportion, especially among women (Fig 4D). Of the 32 women diagnosed from 2015 to 2019, 63% were born in an HIV-endemic country.

The proportion of patients diagnosed with a CD4 count <350 cells/μL could be decreasing over time for different exposure categories (S2 Fig) and gender (Fig 5A) but linear trends were not statistically significant. Overall, the proportion of patients with a CD4 count <200 cells/μL decreased from 16% in 1995 to 4% in 2019 (p-trend = 0.004, $R^2$ = 75%). Negative linear trends in late diagnosis (<200 cells/μL) were also observed among all men (p-trend = 0.004, $R^2$ = 70%, Fig 5B), homosexual men (p-trend = 0.005, $R^2$ = 100%), patients having risky sexual behaviours (p-trend = 0.006, $R^2$ = 100%), patients not using injection drugs (p-trend = 0.007,

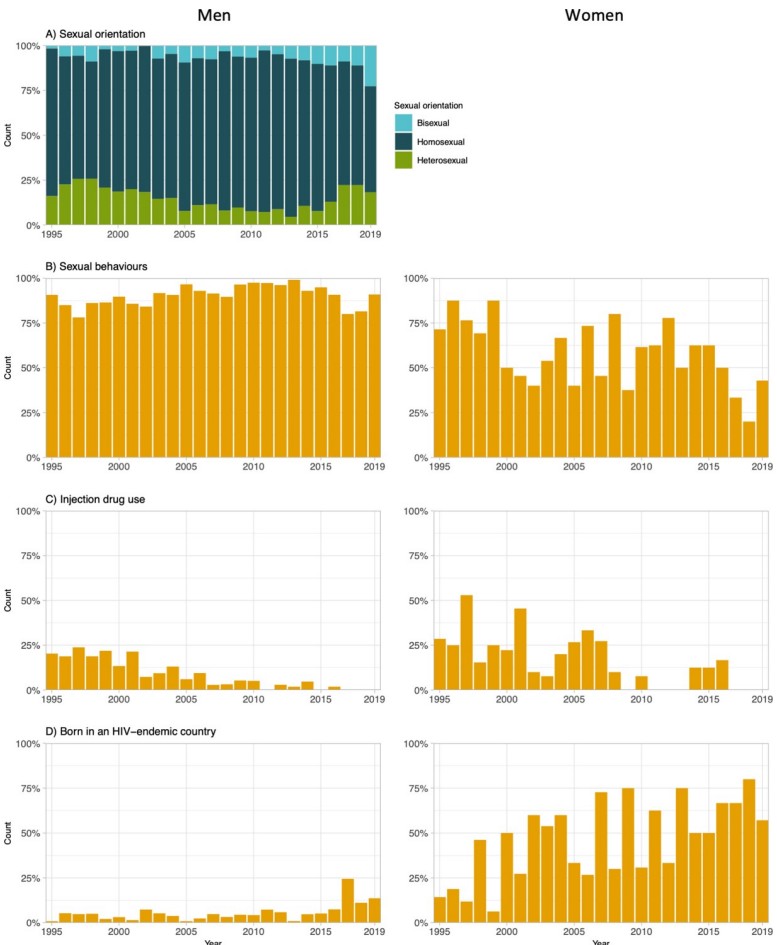

**Fig 4. Prevalence of exposure categories at HIV diagnosis among participants to the HIV-positive cohort at the *Clinique médicale L'Actuel* in Montréal, Canada, over 1995–2019.** For panels A to D, the left and right figures present proportions among all men (n = 2,342) and among all women (n = 270), respectively. Sexual behaviours include condomless sex, having a partner at risk of HIV, having an HIV-positive partner, having multiple sexual partners and/or having ever engaged in sex work.

$R^2$ = 100%), and patients born in an non-endemic country (p-trend = 0.002, $R^2$ = 100%; S3 Fig).

As we observed a shift in new HIV diagnoses towards individuals born in HIV-endemic countries, we compared the characteristics of participants at HIV diagnosis by origin and gender (Table 2). Among all newly diagnosed participants since 1995, a total of 219 (8%) were born in an HIV-endemic country, of whom 179 reported the name of their country. Of those 179 participants, the great majority were born in a sub-Saharan African country (59%) or in Haïti (30%). Men from HIV-endemic countries were mostly heterosexual (75%, 80/106) contrary to men from non-endemic countries who mostly reported being homosexual (83%, 1,852/2,236). While a third (33%, 52/157) of women from a non-endemic country reported injection drug use, none did so among women born in an HIV-endemic country. Both men and women born in HIV-endemic countries tended to have lower CD4 counts at diagnosis than their counterparts from non-endemic countries (Table 2). However, men born in HIV-endemic countries had the lowest median CD4 count of all (285 cells/µL; IQR = 158–463 cells/µL, n = 72) and the higher proportions of late diagnoses with 59% and 37% of them being diagnosed with CD4 counts <350 cells/µL and <200 cells/µL, respectively.

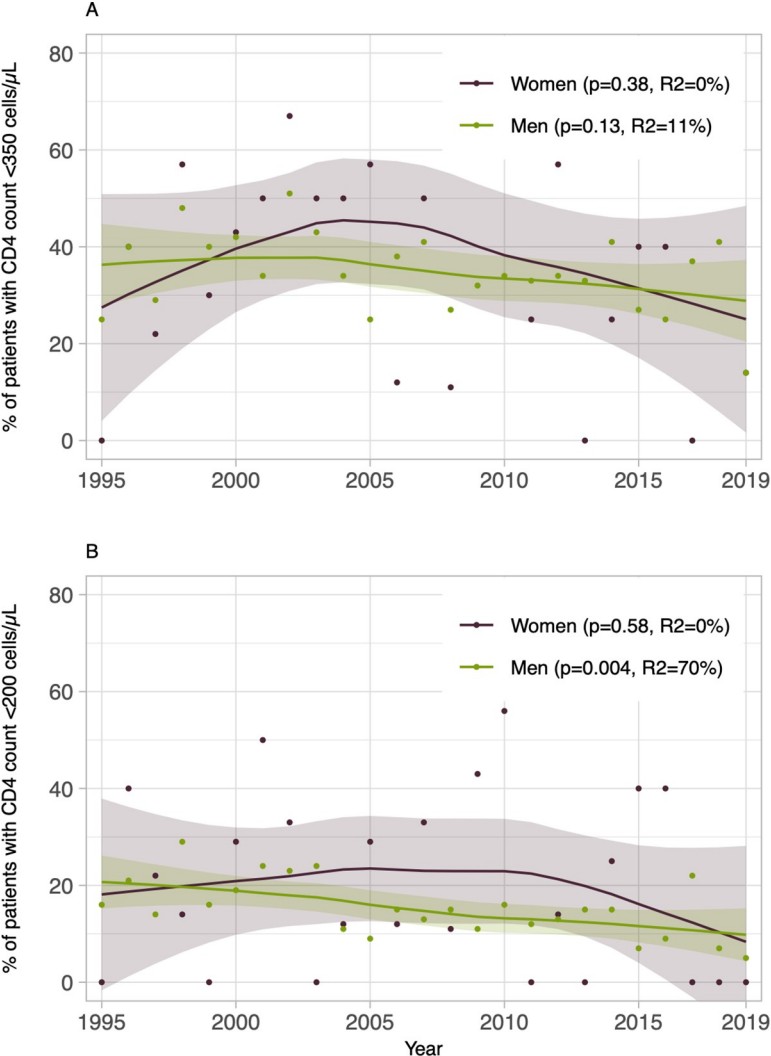

**Fig 5. Trends in the proportion of patients with late or advanced stage of HIV infection at diagnosis among participants to the HIV-positive cohort at the** *Clinique médicale l'Actuel* **in Montréal, Canada, stratified by sex (n = 1,890), 1995–2019.** Late stage of HIV infection (A) is defined as a CD4 count <350 cells/μL, and an advanced stage (B), as a CD4 count <200 cells/μL. Trends are displayed using local linear fitting and a degree of smoothing of 0.75. Time trends (p-values) were assessed by the mean of t-tests within univariate meta-regressions where the year of diagnosis was input as a continuous predictor variable.

We compared the characteristics of participants with and without CD4 count at diagnosis in a post-hoc sensitivity analysis (S2 Table). Compared to the group of participants with missing data for the CD4 count at diagnosis, the group without missing data tended to be older, had slightly higher proportions of men, MSM, and exposure through sexual behaviours, and lower proportions of Black patients, PWID, and patients born in an HIV-endemic country.

## Discussion

HIV elimination efforts should be based on a granular understanding of shifts in epidemiology of this virus. Using clinical data spanning 25 years, this study highlights the changing portrait of the epidemic. First, the proportion of individuals belonging to the MSM exposure category, along with those with an history of injection drug use, decreased through time. Second, people

**Table 2. Characteristics of patients at HIV diagnosis among participants to the HIV-positive cohort at the *Clinique médicale l'Actuel* in Montréal (Canada), stratified by origin and gender (n = 2,612), 1995–2019.**

| | Born in a non-endemic country | | | | | | Born in an HIV-endemic country | | | | | |
|---|---|---|---|---|---|---|---|---|---|---|---|---|
| | Overall | | Men | | Women | | Overall | | Men | | Women [a] | |
| | n = 2,393 | | n = 2,236 | | n = 157 | | n = 219 | | n = 106 | | n = 113 | |
| | n | (%) | n | (%) | n | (%) | n | (%) | n | (%) | n | (%) |
| Mean age, years (SD) | 36 | (10) | 36 | (10) | 34 | (11) | 34 | (10) | 35 | (10) | 33 | (10) |
| Ethno-cultural background | | | | | | | | | | | | |
| Caucasian | 956 | (40) | 907 | (41) | 49 | (31) | 4 | (2) | 2 | (2) | 2 | (2) |
| Black | 46 | (2) | 36 | (2) | 10 | (6) | 198 | (90) | 93 | (88) | 105 | (93) |
| Hispanic | 175 | (7) | 168 | (8) | 7 | (5) | 8 | (4) | 7 | (7) | 1 | (1) |
| Asian | 65 | (3) | 64 | (3) | 1 | (0.6) | 1 | (0.5) | 1 | (1) | 0 | (0) |
| Native | 9 | (0.4) | 8 | (0.4) | 1 | (0.6) | 0 | (0) | 0 | (0.0) | 0 | (0) |
| Other | 48 | (2) | 48 | (2) | 0 | (0) | 3 | (1) | 0 | (0.0) | 3 | (3) |
| Unknown | 1,094 | (46) | 1,105 | (47) | 89 | (57) | 5 | (2) | 3 | (3) | 2 | (2) |
| Sexual orientation | | | | | | | | | | | | |
| Homosexual | 1,853 | (77) | 1,852 | (83) | 1 | (0.6) | 23 | (11) | 22 | (21) | 1 | (1) |
| Heterosexual | 398 | (17) | 248 | (11) | 150 | (96) | 192 | (88) | 80 | (75) | 112 | (99) |
| Bisexual | 142 | (6) | 136 | (6) | 6 | (4) | 4 | (2) | 4 | (4) | 0 | (0) |
| Exposure category [b] | | | | | | | | | | | | |
| Sexual behaviours | 2,214 | (93) | 2,085 | (93) | 129 | (82) | 79 | (36) | 44 | (42) | 35 | (31) |
| Condomless sex | 1,415 | (59) | 1,364 | (61) | 51 | (33) | 34 | (16) | 20 | (19) | 14 | (12) |
| Having a partner at risk of HIV [c] | 735 | (31) | 653 | (29) | 82 | (52) | 49 | (22) | 24 | (23) | 25 | (22) |
| Having an HIV-positive partner | 477 | (20) | 425 | (19) | 52 | (33) | 29 | (13) | 13 | (12) | 16 | (14) |
| Having multiple sexual partners (>1) | 671 | (28) | 654 | (29) | 17 | (11) | 9 | (4) | 9 | (9) | 0 | (0) |
| Having ever engaged in sex work | 72 | (3) | 49 | (2) | 23 | (15) | 1 | (0.5) | 1 | (1) | 0 | (0) |
| Injection drug use | 270 | (11) | 218 | (10) | 52 | (33) | 1 | (0.5) | 1 | (1) | 0 | (0) |
| Contaminated blood transfusion | 4 | (0.2) | 3 | (0.1) | 1 | (0.6) | 2 | (1) | 1 | (1) | 1 | (1) |
| Other [d] | 8 | (0.3) | 6 | (0.3) | 2 | (1) | 2 | (1) | 2 | (2) | 0 | (0) |
| CD4 count, cells/μL (n = 1,890) | | | | | | | | | | | | |
| Median (IQR) | 445 | (290–620) | 447 | (290–620) | 430 | (280–600) | 315 | (168–502) | 285 | (158–463) | 392 | (198–545) |
| ≥500 | 745 | (42) | 706 | (42) | 39 | (42) | 34 | (26) | 16 | (22) | 18 | (30) |
| 350–499 | 419 | (24) | 398 | (24) | 21 | (23) | 28 | (21) | 13 | (18) | 15 | (25) |
| 200–349 | 341 | (19) | 324 | (19) | 17 | (18) | 28 | (21) | 16 | (22) | 12 | (20) |
| 50–199 | 191 | (11) | 177 | (11) | 14 | (15) | 32 | (24) | 21 | (29) | 11 | (18) |
| <50 | 62 | (4) | 60 | (4) | 2 | (2) | 10 | (8) | 6 | (8) | 4 | (7) |

SD: Standard deviation, IQR: Interquartile range.

[a] Women include one transwoman who reported being homosexual and to have sex with men.

[b] Categories are not mutually exclusive.

[c] Partner at risk of HIV includes people who inject drugs, HIV-positive individuals, individuals born in an HIV-endemic country, bisexuals, sex workers, and hemophiliacs.

[d] Other includes accidental exposure and vertical acquisition.

born in HIV-endemic countries now constitute an increasing share of all new diagnoses, especially among women. Third, they tend to be diagnosed at later stages of HIV infection.

In Canada, MSM have been historically disproportionally affected by HIV. Our results suggest that, although MSM always accounted for the highest proportion of new diagnoses, their absolute number and proportion are decreasing since 2013. This trend is consistent with the decrease in the number of new diagnoses among MSM reported at the provincial level, and the

decrease observed in the proportion of gay, bisexual and other MSM among newly diagnosed individuals in the rest of Canada [3, 6]. Such decrease could potentially be explained at least in part by the scale-up of antiretroviral therapy following the World Health Organization's recommendation towards earlier treatment initiation, and by the introduction of PrEP in Montréal [12], fostering access to and uptake of PrEP, especially among MSM [13–15].

We also observed a decline in the proportion of newly diagnosed PWID in our study, especially since the early 2000s. A previous longitudinal study among PWID in Montréal has shown an annual HIV incidence decline of 0.06 cases/100 person-years prior to 2000 followed by a 4-fold more rapid annual decline from 2000 onward [17]. These reductions in proportion of PWID and in HIV incidence may be largely related to changes and enhancements in HIV prevention in the late 1990's. Montréal was one of the firsts North American cities to implement needle and syringe exchange programs to prevent sharing of injection materials, an important factor in HIV transmission among PWID [17]. Its yearly distribution of syringes greatly increased from 340,000 syringes in 1996 to 950,000 syringes in 1999 [17]. According to the SurvUDI, a provincial bio-behavioral surveillance network for HIV infection among PWID, the proportion of PWID reporting sharing needles decreased by 27 percentage-points between 1995 and 2016, while diagnosis and antiretroviral therapy coverage increased substantially between 2003 and 2016 within this population [18].

The decrease in the proportion of MSM and PWID was accompanied by an important increase in the proportion of newly diagnosed individuals from HIV-endemic countries. This exposure group was characterized mostly by heterosexual transmission and a higher representation of women compared to people born in non-endemic countries. Noticeably, people born in HIV-endemic countries had lower CD4 count at diagnosis and were more likely to get diagnosed with a CD4 count <350 cells/μL or <200 cells/μL. The last two metrics seemed to decrease over time among both patients from endemic and from non-endemic countries, suggesting more rapid diagnosis. However, patients from endemic countries had consistently higher proportions of new diagnoses at late or advanced stages of HIV infection, suggesting persistent longer delays to diagnosis or poor linkage to antiretroviral therapy among this group as compared to others. A recent study has shown that, among newly diagnosed asylum seekers in a primary HIV referral centre in Montréal between 2017 and 2018, the median time from entry into Canada to HIV testing was estimated at 27 days, almost two-thirds of patients were late presenters (CD4 <350 cells/μL), almost a quarter presented with an advanced stage of HIV infection (CD4 <200 cells/μL), and only 45% of patients were linked to care within 30 days [19, 20]. In Canada, people born in HIV-endemic countries are disproportionately affected by social and economic factors that increase their vulnerability to HIV infection and act as barriers to prevention, diagnosis and treatment programs [21]. Attention should be paid to adapt programs for people from HIV-endemic countries who faces distinct barriers to rapid diagnosis [19, 22]. Improving provision of information at the time of entry into Canada about the availability of free HIV-testing services and medical care, and about the absence of denial of entry or residency in Canada based on a positive HIV diagnosis could help in increasing and accelerating access to HIV screening and treatment among migrants from HIV-endemic countries [19]. To better inform public health, additional studies are required to describe if patients from endemic countries are more likely to have been infected before their entry to Canada or to acquire the infection in Canada due to unfavorable socioeconomic factors.

Our findings must be interpreted in light of inherent limitations. First, we used data from a single but large clinic in Montréal that represented a convenience sample of patients. However, the shift we observed in exposure categories are consistent with those observed in Québec and in Canada [3, 6]. Second, some of our subgroup analyses relied on small number of HIV diagnoses that could have led to imprecise estimates. Third, we retrospectively included patients

diagnosed before 1999 and this could have resulted in a selection bias if patients with poorer prognosis had died before consenting participation. To alleviate this, we restricted our analyses to the ART era. Fourth, there was a high amount of missing data for the CD4 count at diagnosis leading to high variability of estimates. Fifth, we cannot distinguish the country where HIV was acquired for people born in an endemic country–important information for prevention programs. Finally, most of the variables were self-reported. Potential misreporting, especially if differential between the subgroups being compared, could have biased our inferences (e.g. reporting of sexual orientation among men born in HIV-endemic vs non-endemic countries).

Strengths of this study include the analysis of cohort data from a large sexual health clinic specialized in the diagnosis and treatment of STBBI–cohort spanning 25 years. This cohort provides a unique opportunity to understand how the characteristics of diagnosed PLHIV evolved during the ART era and to identify changing needs in HIV prevention and treatment and care cascade.

## Conclusions

In conclusion, MSM continue to account for the largest exposure category in absolute numbers. However, patients born in HIV-endemic countries constitute an increasing proportion of newly diagnosed individuals. Tailoring HIV testing strategies and other prevention interventions to these individuals is warranted.

## Supporting information

**S1 Fig. Trends in mean age at diagnosis by exposure category, 1995–2019 (n = 2,621).** Panel A presents trends by sexual orientation group among men only (n = 2,342). Panels B to D present trends among combined men and women by sexual risk behaviours (including condomless sex, having a partner at risk, having an HIV-positive partner, having multiple sexual partners and/or having ever engaged in sex work), injection drug use, and origin (being born in an HIV-endemic country or not), respectively. Trends are displayed using local linear fitting and a degree of smoothing of 0.75. Time trends (p-values) were assessed by the mean of t-tests within univariate meta-regressions where the year of diagnosis was input as a continuous predictor variable.
(TIF)

**S2 Fig. Trends in the proportion of patients with late stage of HIV infection (<350 CD4 cells/µL) at diagnosis among participants to the HIV-positive cohort at the *Clinique médicale l'Actuel* in Montréal, Canada, stratified by exposure category (n = 1,890), 1995–2019.** Panel A presents trends by sexual orientation group among men only (n = 1,737). Panels B to D present trends among combined men and women by sexual risk behaviours (including condomless sex, having a partner at risk, having an HIV-positive partner, having multiple sexual partners and/or having ever engaged in sex work), injection drug use, and origin (being born in an HIV-endemic country or not), respectively. Trends are displayed using local linear fitting and a degree of smoothing of 0.75. Time trends (p-values) were assessed by the mean of t-tests within univariate metaregressions where the year of diagnosis was input as a continuous predictor variable.
(TIF)

**S3 Fig. Trends in the proportion of patients with advanced stage of HIV infection (<200 CD4 cells/µL) at diagnosis among participants to the HIV-positive cohort at the *Clinique médicale l'Actuel* in Montréal, Canada, stratified by exposure category (n = 1,890), 1995–2019.** Panel A presents trends by sexual orientation group among men only (n = 1,737). Panels

B to D present trends among combined men and women by sexual risk behaviours (including condomless sex, having a partner at risk, having an HIV-positive partner, having multiple sexual partners and/or having ever engaged in sex work), injection drug use, and origin (being born in an HIV-endemic country or not), respectively. Trends are displayed using local linear fitting and a degree of smoothing of 0.75. Time trends (p-values) were assessed by the mean of t-tests within univariate metaregressions where the year of diagnosis was input as a continuous predictor variable.
(TIF)

**S1 Table. Checklist of items, extended from the STROBE statement, that should be reported in observational studies using routinely collected health data (RECORD).** (PDF)

**S2 Table. Characteristics of patients at HIV diagnosis among participants to the HIV-positive cohort at the *Clinique médicale l'Actuel* in Montréal (Canada), stratified by availability of CD4 count at diagnosis (n = 2,612), 1995–2019.** (PDF)

## Acknowledgments

We thank all participants to the cohort of HIV-positive patients from the *Clinique médicale l'Actuel*, as well as all the clinicians who practiced at the clinic since its opening in 1987, without whom this study could not have taken place. Special thanks also to the clinic's Director of Development and Strategic Planification, Anne-Fanny Vassal, for her logistical support and coordination with clinicians.

## Author Contributions

**Conceptualization:** Katia Giguère, Mathieu Maheu-Giroux.

**Data curation:** Maliheh Vaziri.

**Formal analysis:** Katia Giguère.

**Funding acquisition:** Katia Giguère, Mathieu Maheu-Giroux.

**Investigation:** Katia Giguère.

**Methodology:** Katia Giguère, Mathieu Maheu-Giroux.

**Supervision:** Mathieu Maheu-Giroux.

**Validation:** Maliheh Vaziri, Clément Olivier, Louise Charest, Jason Szabo, Réjean Thomas.

**Writing – original draft:** Katia Giguère.

**Writing – review & editing:** Clément Olivier, Louise Charest, Jason Szabo, Réjean Thomas, Mathieu Maheu-Giroux.

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
