## [Decision Letter · Decision Letter 0]

11 Jun 2021

PONE-D-21-13938

Characteristics of new HIV diagnoses over 1983-2019: a clinic-based study in Montréal, Canada

PLOS ONE

Dear Dr. Giguère,

Thank you for submitting your manuscript to PLOS ONE. After careful consideration, we feel that it has merit but does not fully meet PLOS ONE’s publication criteria as it currently stands. Specifically please take note of and respond to the reviewers' comments, especially that of Reviewer 2, in order to improve the scientific rigor  and therefore reliability of the results. Therefore, we invite you to submit a revised version of the manuscript that addresses the points raised during the review process. Kindly be advised that acceptance of the revised manuscript is not guaranteed.

We look forward to receiving your revised manuscript.

Kind regards,

Shui Shan Lee

Academic Editor

PLOS ONE

Additional Editor Comments (if provided):

This is an epidemiologic study founded on the analyses of routinely collected surveillance data over a long period of 40 years. The manuscript contains data and information which could be of useful reference to epidemiologists, and could strengthen the knowledgebase of HIV epidemiology in North America. There are however some methodological flaws which have been raised by one of the reviewers. Major revisions are needed before this manuscript could be considered for publication.

Journal Requirements:

2)  Thank you for stating the following in the Competing Interests section:

[I have read the journal's policy and the authors of this manuscript have the following

competing interests: KG reports a postdoctoral award from the Fonds de recherche du

Québec - Santé, during the conduct of the study, and personal fees from UNAIDS,

outside the submitted work; MMG’s research program is supported by a Canada

Research Chair (Tier 2) in Population Health Modeling, and grants from the Canadian

Foundation for AIDS Research and the Canadian Institutes of Health Research, and

funding from UNAIDS and WHO, outside the submitted work.].

3) We note that you have indicated that data from this study are available upon request. PLOS only allows data to be available upon request if there are legal or ethical restrictions on sharing data publicly. For information on unacceptable data access restrictions, please see http://journals.plos.org/plosone/s/data-availability#loc-unacceptable-data-access-restrictions.

Reviewers' comments:

Reviewer's Responses to Questions

**Comments to the Author**

1. Is the manuscript technically sound, and do the data support the conclusions?

Reviewer #1: Yes

Reviewer #2: No

2. Has the statistical analysis been performed appropriately and rigorously? 

Reviewer #1: Yes

Reviewer #2: No

3. Have the authors made all data underlying the findings in their manuscript fully available?

Reviewer #1: No

Reviewer #2: Yes

4. Is the manuscript presented in an intelligible fashion and written in standard English?

Reviewer #1: Yes

Reviewer #2: Yes

5. Review Comments to the Author

Reviewer #1: - The authors mentioned there are some restrictions on data availability but details are not specified.

- The authors provided a comprehensive analysis on the characteristics of newly diagnosed HIV patients in their center over nearly 4 decades and identified the changes in their trends including the exposure risk. This information is important to inform the health authority to tailor the testing and prevention strategies for the control of HIV transmission.

- The authors reported the decreasing trend of mean CD4 at diagnosis from 1381 to 590 cells/uL over the study period though not statistically significant. This may imply the patients presented later to the service over the years. Any reasons observed to explain this finding?

- Is there any reason why the baseline HIV viral load data is not included in the analysis? The change in the baseline HIV viral load over the study period may have implications on the disease transmission in the city.

- The authors found an important increase in the proportion of patients born in an HIV endemic country. As pointed out by the authors, it would be important to know if they are infected before their entry to Canada or acquire the infection in Canada due to the unfavorable socioeconomic factors rendering them at higher risk of infection. This should be further studied to improve the diagnosis and prevention programs.

Reviewer #2: Giguère and colleagues have performed a statistical analysis on new HIV diagnosis in a clinical setting in Montréal, Canada. The authors had access to data from 1999 onwards including individuals that had been diagnosed with HIV as early as 1983. The authors conclude that although men-who-have-sex-with-men (MSM) remain the predominant route of transmission, the proportion of individuals from HIV endemic countries is increasing. The authors suggest that individuals from HIV endemic countries could face distinct barriers to

rapid diagnosis. The paper also presents data showing that from 1983 to 2019, the mean age at diagnosis increased from 26 to 31 years (p-trend <0.0001) and the mean CD4 count at diagnosis decreased from 1,381 to 590 cells/μL (not significantly). I have several concerns about the paper.

Major comments

1. The authors show an impact of calendar time (1983 to 2019) on age at diagnosis and CD4 cell count. Data used in the paper were collected starting in 1999. Patients that were diagnosed before combination antiretroviral therapy became available in the mid-1990s had a very high risk of mortality (especially the ones diagnosed in the 1980s). As the authors acknowledge in the discussion, inclusion of these patients diagnosed well before effective treatment became available, are likely to have passed away and not have been available for inclusion by 1999. Also, the number of new HIV diagnoses was lower in the early years of the pandemic. Inclusion of data before 1999 will therefore result in a bias in the statistical analysis, as the ones who survived in the period before effective treatment became available are likely to have had a better immune response to HIV (in my view this is also illustrated by the high CD4 cell count at diagnosis found in 1983 and maybe even the younger age at diagnosis in 1983). The paper should therefore be restricted to patients diagnosed from 1999 onwards. Prevention gaps required to meet WHO’s 90-90-90 goals also depend on patients diagnosed in more recent years and not in the time before treatment became available.

2. The CD4 cell count is presented using a mean and standard-deviation assuming a normal distribution. The CD4 cell count is, however, not normally distributed (see for instance this paper: https://aidsrestherapy.biomedcentral.com/articles/10.1186/1742-6405-8-35) . The CD4 cell count should therefore be represented as a median and a type of range. The mean CD4 cell count should not be considered in the statistical analysis.

3. The CD4 cell count is highly variable within and between individuals. In the literature, there is a lot of emphasis on diagnosis in a late stage (CD4<350) or an advanced stage of infection (CD4<200). Unfortunately, the paper does not present the proportion of patients diagnosed in a more advanced stage of infection. The statement in the abstract that people from non-endemic countries are diagnosed at a mean CD4 cell count of 484 and from endemic countries at a CD4 of 374 is therefore not so relevant if all these individuals were diagnosed at a CD4 > 350. I recommend that the authors present the proportion of people diagnosed late and in an advanced stage of infection. A change over time in these proportions would be very relevant information, especially in the most recent years.

4. The introduction covers three pages which in my view is very long. The introduction also includes a lot of historical information about HIV in Quebec that is not very relevant. I recommend to strongly shorten the introduction.

5. Figure 3 shows the fit of regression on the mean age at diagnosis by gender since 1983. Did the authors investigate if the fit improved by using a more complex regression equation as compared to a simpler model?

Minor

1. Please improve the resolution of figure 4.

6. PLOS authors have the option to publish the peer review history of their article (what does this mean?). If published, this will include your full peer review and any attached files.

Reviewer #1: No

Reviewer #2: No

---

## [Author Response · Author response to Decision Letter 0]

4 Aug 2021

Editor's Comments 

1) Please ensure that your manuscript meets PLOS ONE's style requirements, including those for file naming. The PLOS ONE style templates can be found at https://journals.plos.org/plosone/s/file?id=wjVg/PLOSOne_formatting_sample_main_body.pdf and https://journals.plos.org/plosone/s/file?id=ba62/PLOSOne_formatting_sample_title_authors_affiliations.pdf

Response. We have carefully reviewed the manuscript, tables and figures to ensure that they meet PLOS ONE's style requirements. Changes and corrections include the use of Level 2 heading (bold type, 16 pt font) for sub-sections of major sections and use of double-space paragraph format for the entire manuscript, including the captions.

2) Thank you for stating the following in the Competing Interests section: 

[I have read the journal's policy and the authors of this manuscript have the following competing interests: KG reports a postdoctoral award from the Fonds de recherche du Québec - Santé, during the conduct of the study, and personal fees from UNAIDS, outside the submitted work; MMG’s research program is supported by a Canada Research Chair (Tier 2) in Population Health Modeling, and grants from the Canadian Foundation for AIDS Research and the Canadian Institutes of Health Research, and funding from UNAIDS and WHO, outside the submitted work.]. 

Response. We have updated the Competing Interests as follows:

"I have read the journal's policy and the authors of this manuscript have the following competing interests: KG reports postdoctoral awards from the Fonds de recherche du Québec - Santé, the Canadian Institutes of Health Research (176645), and the CIHR Canadian HIV Trials Network during the conduct of the study, and personal fees from UNAIDS, outside the submitted work; MMG’s research program is supported by a Canada Research Chair (Tier 2) in Population Health Modeling, and grants from the Canadian Foundation for AIDS Research and the Canadian Institutes of Health Research, and funding from UNAIDS and WHO, outside the submitted work. This does not alter our adherence to PLOS ONE policies on sharing data and material."

There are restrictions on sharing of data and/or materials and we have stated them as follows with the initial submission of our manuscript to PLOS ONE:

"Data cannot be shared publicly because they contain potentially identifying or sensitive patient information. The data underlying the results presented in the study are available from the Clinique médicale l'Actuel (https://cliniquelactuel.com/)."

Response. We have included our updated Competing Interests statement to our cover letter.

3) We note that you have indicated that data from this study are available upon request. PLOS only allows data to be available upon request if there are legal or ethical restrictions on sharing data publicly. For information on unacceptable data access restrictions, please see http://journals.plos.org/plosone/s/data-availability#locunacceptable- data-access-restrictions. 

Response. As mentioned above, there are indeed restrictions on sharing of data and we have explained them:

"Data cannot be shared publicly because they contain potentially identifying or sensitive patient information. The data underlying the results presented in the study are available from the Clinique médicale l'Actuel (https://cliniquelactuel.com/)."

If anything is still missing or not properly reported, kindly let us know exactly what and we will be pleased to update this declaration accordingly.

b) If there are no restrictions, please upload the minimal anonymized data set necessary to replicate your study findings as either Supporting Information files or to a stable, public repository and provide us with the relevant URLs, DOIs, or accession numbers. Please see http://www.bmj.com/content/340/bmj.c181.long for guidelines on how to deidentify and prepare clinical data for publication. For a list of acceptable repositories, please see http://journals.plos.org/plosone/s/data-availability#loc-recommended-repositories. We will update your Data Availability statement on your behalf to reflect the information you provide.

Response. Not applicable as there are ethical restrictions that prohibits us to share the data.

Reviewer #1 

1. The authors mentioned there are some restrictions on data availability but details are not specified.

Response. As mentioned by the reviewer, we indeed reported restrictions on data availability. However, contrary to what is mentioned by the reviewer, we had also reported details on these restrictions as follows:

"Data cannot be shared publicly because they contain potentially identifying or sensitive patient information. The data underlying the results presented in the study are available from the Clinique médicale l'Actuel (https://cliniquelactuel.com/)."

2. The authors provided a comprehensive analysis on the characteristics of newly diagnosed HIV patients in their center over nearly 4 decades and identified the changes in their trends including the exposure risk. This information is important to inform the health authority to tailor the testing and prevention strategies for the control of HIV transmission.

Response. We are pleased that the reviewer found that our study results are important to inform the public health response to HIV.

3. The authors reported the decreasing trend of mean CD4 at diagnosis from 1381 to 590 cells/uL over the study period though not statistically significant. This may imply the patients presented later to the service over the years. Any reasons observed to explain this finding?

Response. As per a request of reviewer #2, we do not present trends in mean CD4 count at diagnosis anymore (see response to comments 2 and 3 of reviewer #2 for details). We rather present trends in the proportion of patients with CD4 <350 cells/µL or CD4 <200 cells/µL as proxies for late or advanced stages of HIV infection at diagnosis, respectively. Moreover, we restricted analyses to the ART era (also as requested by reviewer #2 in their first comment) and now present time trends from 1995 to 2019 instead of from 1983 to 2019. We do not observe results suggesting that patients would present later to HIV testing services over the years. Moreover, the trends in proportion of patients with late or advanced stages of HIV infection rather suggest that patients were diagnosed sooner over the years. Unfortunately, available data do not allow us to further investigate the reasons that could explain this finding.

4. Is there any reason why the baseline HIV viral load data is not included in the analysis? The change in the baseline HIV viral load over the study period may have implications on the disease transmission in the city.

Response. HIV viral loads were not routinely collected at diagnosis. That is, only a minority of included patients (13%) had a viral load at diagnosis. Among patients who had a first viral load within a year of HIV diagnosis (72% of analysed patients), time to viral load varied a lot and went from 0 to 365 days, with a median of 26 days and a mean of 53 days. As such, we believe that for many patients, the first viral load was performed after initiation of HIV treatment. Since HIV treatment rapidly decreases viral load, we believe that the first HIV viral load performed within participants would not be a good proxy for viral load at diagnosis and decided not to include this metric in the analyses.

5. The authors found an important increase in the proportion of patients born in an HIV endemic country. As pointed out by the authors, it would be important to know if they are infected before their entry to Canada or acquire the infection in Canada due to the unfavorable socioeconomic factors rendering them at higher risk of infection. This should be further studied to improve the diagnosis and prevention programs.

Response. We agree with the reviewer that understanding if patients from endemic countries were either infected before their entry to Canada or acquired the infection after is important. However, the exact date of infection is rarely known and was not available/reported in our study. We thus added a sentence in the Discussion section to emphasize the reviewer’s point:

"To better inform public health, additional studies are required to describe if patients from endemic countries are more likely to have been infected before their entry to Canada or to acquire the infection in Canada due to unfavorable socioeconomic factors."

Reviewer #2

Giguère and colleagues have performed a statistical analysis on new HIV diagnosis in a clinical setting in Montréal, Canada. The authors had access to data from 1999 onwards including individuals that had been diagnosed with HIV as early as 1983. The authors conclude that although men-who-have-sex-with-men (MSM) remain the predominant route of transmission, the proportion of individuals from HIV endemic countries is increasing. The authors suggest that individuals from HIV endemic countries could face distinct barriers to rapid diagnosis. The paper also presents data showing that from 1983 to 2019, the mean age at diagnosis increased from 26 to 31 years (p-trend <0.0001) and the mean CD4 count at diagnosis decreased from 1,381 to 590 cells/μL (not significantly). I have several concerns about the paper.

Major comments 

1. The authors show an impact of calendar time (1983 to 2019) on age at diagnosis and CD4 cell count. Data used in the paper were collected starting in 1999. Patients that were diagnosed before combination antiretroviral therapy became available in the mid-1990s had a very high risk of mortality (especially the ones diagnosed in the 1980s). As the authors acknowledge in the discussion, inclusion of these patients diagnosed well before effective treatment became available, are likely to have passed away and not have been available for inclusion by 1999. Also, the number of new HIV diagnoses was lower in the early years of the pandemic. Inclusion of data before 1999 will therefore result in a bias in the statistical analysis, as the ones who survived in the period before effective treatment became available are likely to have had a better immune response to HIV (in my view this is also illustrated by the high CD4 cell count at diagnosis found in 1983 and maybe even the younger age at diagnosis in 1983). The paper should therefore be restricted to patients diagnosed from 1999 onwards. Prevention gaps required to meet WHO’s 90-90-90 goals also depend on patients diagnosed in more recent years and not in the time before treatment became available.

Response. We agree with the reviewer that patients who were diagnosed before ART era in the mid-1990s may have experienced higher rates of mortality, preventing them to be recruited to the study from 1999 onwards, and potentially leading to selection bias. We thus restricted analyses to the ART era, i.e. to patients diagnosed from 1995 onwards (and still excluding patients diagnosed in 2020 to alleviate the right truncation). All results, as well as all sections of the manuscript have been updated to take this change into account. In the Methods sections, we have added the underlined sentence below to justify this restriction:

"Because we included both patients diagnosed at l'Actuel and patients initiating care at the clinic but diagnosed elsewhere, we expected that the last few years of data would be affected by right truncation, especially in 2020 due to the COVID-19 global pandemic. In addition, because patients diagnosed before the beginning of ART era, in mid-1990's, could have passed away before recruitment to the cohort in 1999, left truncation could be an issue. To avoid these, analyses were restricted to patients diagnosed between 1995 and 2019."

2. The CD4 cell count is presented using a mean and standard-deviation assuming a normal distribution. The CD4 cell count is, however, not normally distributed (see for instance this paper: https://aidsrestherapy. biomedcentral.com/articles/10.1186/1742-6405-8-35) . The CD4 cell count should therefore be represented as a median and a type of range. The mean CD4 cell count should not be considered in the statistical analysis.

Response. We have replaced mean CD4 counts and SD with median CD4 counts and interquartile ranges, respectively, everywhere in the manuscript. To avoid methodological challenges with the meta-regression of medians, we assessed trends in the proportion of patients with late (CD4 cells count <350 cells/µL) or advanced stage of HIV infection (CD4 cells count <200 cells/µL) at diagnosis. These changes did not alter our conclusions, however. We still do not observe differences in CD4 count at diagnosis between men and women overall, both men and women born in HIV-endemic countries tended to have lower CD4 counts at diagnosis than their counterparts from non-endemic countries, and men born in HIV-endemic countries had the lowest CD4 count of all.

3. The CD4 cell count is highly variable within and between individuals. In the literature, there is a lot of emphasis on diagnosis in a late stage (CD4<350) or an advanced stage of infection (CD4<200). Unfortunately, the paper does not present the proportion of patients diagnosed in a more advanced stage of infection. The statement in the abstract that people from non-endemic countries are diagnosed at a mean CD4 cell count of 484 and from endemic countries at a CD4 of 374 is therefore not so relevant if all these individuals were diagnosed at a CD4 > 350. I recommend that the authors present the proportion of people diagnosed late and in an advanced stage of infection. A change over time in these proportions would be very relevant information, especially in the most recent years.

Response. We thank the reviewer for these relevant suggestions. We added the counts and proportions of patients by CD4 cells count categories at diagnosis (≥500, 350-499, 200-349, 50-199, and <50 cells/µL) within tables. We also present trends in the proportion of patients with late (CD4 cells count <350 cells/µL) or advanced stage of HIV infection (CD4 cells count <200 cells/µL). Interestingly, although not statistically significant for gender and exposure category strata, the results suggest overall decreases over time in the proportion of patients diagnosed at later stages of HIV infection, suggesting earlier diagnosis over the years. 

4. The introduction covers three pages which in my view is very long. The introduction also includes a lot of historical information about HIV in Quebec that is not very relevant. I recommend to strongly shorten the introduction.

Response. Our study being descriptive and spanning over 25 years, we believe that contextualizing the results is warranted for readers of other jurisdiction. Nonetheless, we have shortened the introduction by removing some secondary information.

5. Figure 3 shows the fit of regression on the mean age at diagnosis by gender since 1983. Did the authors investigate if the fit improved by using a more complex regression equation as compared to a simpler model?

Response. We avoided complex regression models as we worried about overfitting the data. Inspection of model fits suggested that linear time trend models were appropriate to describe patterns and that polynomials or logarithmic were unwarranted.

Minor 

1. Please improve the resolution of figure 4.

Response. We have increased the resolution of Figure 4 from 300 dpi to 398 dpi.

---

## [Decision Letter · Decision Letter 1]

27 Sep 2021

Characteristics of new HIV diagnoses over 1995-2019: a clinic-based study in Montréal, Canada

PONE-D-21-13938R1

Dear Dr. Giguère,

We’re pleased to inform you that your manuscript has been judged scientifically suitable for publication and will be formally accepted for publication once it meets all outstanding technical requirements.

Kind regards,

Shui Shan Lee

Academic Editor

PLOS ONE

Additional Editor Comments (optional):

Reviewers' comments:

Reviewer's Responses to Questions

**Comments to the Author**

1. If the authors have adequately addressed your comments raised in a previous round of review and you feel that this manuscript is now acceptable for publication, you may indicate that here to bypass the “Comments to the Author” section, enter your conflict of interest statement in the “Confidential to Editor” section, and submit your "Accept" recommendation.

Reviewer #1: All comments have been addressed

Reviewer #2: All comments have been addressed

2. Is the manuscript technically sound, and do the data support the conclusions?

Reviewer #1: Yes

Reviewer #2: Yes

3. Has the statistical analysis been performed appropriately and rigorously? 

Reviewer #1: Yes

Reviewer #2: Yes

4. Have the authors made all data underlying the findings in their manuscript fully available?

Reviewer #1: Yes

Reviewer #2: Yes

5. Is the manuscript presented in an intelligible fashion and written in standard English?

Reviewer #1: Yes

Reviewer #2: Yes

6. Review Comments to the Author

Reviewer #1: - My comments in the previous review have been appropriately addressed

- The authors changed the analysis period to 1995-2019 to avoid the selection bias. Based on the reasons discussed in the article, I suppose cART was widely used in the clinic since 1995.

Reviewer #2: The authors have correctly addressed my concerns.

7. PLOS authors have the option to publish the peer review history of their article (what does this mean?). If published, this will include your full peer review and any attached files.

Reviewer #1: No

Reviewer #2: No

---

## [Editor Report · Acceptance letter]

29 Sep 2021

PONE-D-21-13938R1 

Characteristics of new HIV diagnoses over 1995-2019: a clinic-based study in Montréal, Canada 

Dear Dr. Giguère:

I'm pleased to inform you that your manuscript has been deemed suitable for publication in PLOS ONE. Congratulations! Your manuscript is now with our production department. 

Kind regards, 

on behalf of

Professor Shui Shan Lee 

Academic Editor

PLOS ONE